# The increasing trend in the consumption of ultra-processed food products is associated with a diet related to chronic diseases in Colombia—Evidence from national nutrition surveys 2005 and 2015

**Gustavo Cediel**[1]*, **Elisa María Cadena**[2], **Pamela Vallejo**[3], **Diego Gaitán**[1], **Fabio Da Silva Gomes**[4]

**1** Unidad de Problemáticas de Interés en Nutrición Pública, Grupo de Investigación Saberes Alimentarios, Escuela de Nutrición y Dietética, Universidad de Antioquia UdeA, Medellín, Colombia, **2** Faculty of Social Sciences, Department of Psychology, University of the Andes, Bogotá, Colombia, **3** Ministry of Health and Social Protection, Bogotá, Colombia, **4** Pan American Health Organization, Washington, D.C., United States of America

* gustavo.cedielg@udea.edu.co

**Data Availability Statement:** This cross-sectional analysis used data from the first (2005 y) and last National Survey (2015 y) of the Nutritional

## Abstract

### Introduction

Ultra-processed food products (UPF) have been related to chronic diseases (CD). Public health politics has been establishing strategies to decrease the consumption of these products in the country.

### Objectives

i) To assess the trend of the consumption of UPF between 2005 and 2015. (ii) its association with sociodemographic factors and the overall dietary content of nutrients related to CD in 2015. (iii) to estimate the Population Attributable Fraction of unhealthy nutrient intake in Colombia in 2015 due to ultra-processed food consumption.

### Methods

We used data from the first (2005) and the last (2015) National Surveys of the Nutritional Status in Colombia. Food consumption was assessed using a 24-hour food recall. The NOVA classification classified the food items according to the extent and purpose of industrial processing.

### Results

The consumption of processed and UPF increased in Colombia between 2005 and 2015. In 2015, no significant differences were found in the consumption of UPF between men and women but significant differences by age, wealth index, area of residence, and ethnicity (p<0.001). A significant positive association was found between the dietary share of UPF

Situation of Colombia with consumption data collected with a 24-hour recall technique (ENSIN, 2005, 2015) and the National Demographic and Health Survey of Colombia (ENDS, 2005, 2015). ENSIN data were combined with ENDS data to link food consumption with demographic information at the individual level. The size and design of the sample were carried out to calculate the estimates of proportion and prevalence, as well as statistical models to evaluate associations -ICBF. Instituto Colombiano de Bienestar Familiar. ENSIN - Encuesta Nacional de Situación Nutricional en Colombia. Primera Ed. Bogotá, 2005. -Gobierno Nacional. ENSIN: Encuesta Nacional de Situación Nutricional 2015. Instituto Colombiano. Bogotá, 2019https://www.icbf.gov.co/bienestar/nutricion/encuesta-nacional-situacion-nutricional.

**Funding:** This work was supported by the "Vicerrectoria de Investigación de la Universidad de Antioquia" under the project number 2020-35090 where GC is the principal investigator and DG is coinvestigator. The Ministry of health and social protection of Colombia provide the database. The funders had no role in study design, data collection and analysis, decision to publish, or preparation of the manuscript.

**Competing interests:** The authors have declared that no competing interests exist.

and the content of CD-related nutrients such as free sugars, total fats, saturated fats, trans-fats, and sodium. The prevalence of excessive intake of all CD-related nutrients (according to WHO recommendations) increased across quintiles of the dietary share of UPF. With the reduction of UPF consumption to the level seen among the 20% lowest consumers [1.0% (0–4.5%) of the total energy from UPF], the prevalence of excessive nutrient intake was almost three-fourths lower for trans fats; around one third lower for free sugar and saturated fats, 26% lower for sodium and 15% lower for total fat.

## Conclusions

In Colombia, the increasing trend in the consumption of UPF is associated with increasing intake of CD-related nutrients. Thus, reducing the consumption of UPF is a potentially effective way to achieve the nutritional goals of the WHO for the prevention of CD.

## Introduction

Ultra-processed food products (UPF) are industrial formulations made from substances derived from foods or synthesized from other organic sources. Usually, they contain little or no natural foods, high fat, sodium, or sugar content, and low content of dietary fiber, water, protein, micronutrients, and bioactive compounds [1]. These products dominate the food system in developed countries, contributing almost 60% of the total energy consumed in the United States [2] and the United Kingdom [3] and 50% in Canada [4]. The evidence in several countries shows that consumption of UPF generates a dietary pattern related to worsening diet quality and the presence of obesity and chronic diseases (CD) [5]. Given this evidence, efforts to reduce the intake of these products and nutrients related to CD through regulatory policies have been promoted in the region of the Americas [6, 7].

The Pan American Health Organization (PAHO) reported a growing trend in the sales of UPFs in Latin America, with a 26% increase between 2000 and 2013. In Colombia a similar increase of 25% over the same period was observed [8]. Additionally, analysis of food consumption at the national level in 2005 showed that 15.9% of the energy consumed by the population of Colombia came from UPF [9]. The highest consumption was observed among adolescents, with UPF contributing 20.3% to the total daily energy intake [10, 11].

Data from Euromonitor International show an annual percentage growth of more than 6.0% in sales of all types of UPF in Colombia [12]. The relative increase in sales within such product categories was higher in developing countries, including Colombia, than in industrialized countries between 1998 and 2012. Other Latin American countries, such as Brazil, Chile, and Mexico, have found substantial incorporation of UPF in the diets of their populations, with 21.5% in 2008–2009, 28.6% in 2010, and 30% in 2012, respectively [13–15]. These trends suggest that the population of Colombia, as in neighboring countries, is vulnerable to a dramatic increase in UPF consumption.

In concordance, evidence shows how the corporative food regime has been established in Colombia since 1990 [16, 17]. In the context of increasing foreign investment of food manufacturing companies, expanding the presence of UPF in diets and food systems, and growing trends in obesity and CD [18], it is crucial to know the pattern and direction of consumption of these UPF in Colombia. Due to this background, the objectives of this work are: i) to assess the trend consumption of ultra-processed foods between 2005 and 2015, (ii) its

association with sociodemographic factors and the overall dietary content of nutrients related to CD in 2015, (iii) and to estimate the Population Attributable Fraction of unhealthy nutrient intake in Colombia in 2015 due to ultra-processed food consumption.

## Methods

### Data source

This cross-sectional analysis used secondary data from the first (2005) and last National Survey (2015) of the Nutritional Situation of Colombia with consumption data collected with a 24-hour recall technique (ENSIN 2005 and 2015) and the National Demographic and Health Survey of Colombia (ENDS, 2005 and 2015). ENSIN data were combined with ENDS data to link food consumption with demographic information at the individual level. The size and design of the sample were carried out to calculate the estimates of proportion and prevalence, as well as statistical models to evaluate associations [19, 20].

The dietary data within ENSIN were obtained using one 24-hour recall, administered by the interviewer, in ages between age >1y and 64 years for both sexes. The 24-hour recall captured data from random weekdays and weekends. Interviewers used standardized food models to improve the accuracy of the amount and weight of food and beverages consumed. Information on the food type, the preparation's name, ingredients, and the amount consumed were recorded. The person responsible for preparing the food was present during the interview. In the cases in which the food consumed by a child was at school or in a daycare center, the interviewer visited the school to obtain detailed information on the preparations. The data quality was controlled throughout the process, and the interview was repeated in case of inconsistencies [19, 20].

### Food classification according to processing

Food items from the two surveys were classified into one of four NOVA categories. The categories are mutually exclusive and vary depending on their extent and purpose of processing. They include 1) unprocessed or minimally processed foods, 2) processed culinary ingredients, 3) processed foods, and 4) ultra-processed products [1, 21]. Foods were categorized into one of 33 subgroups within the NOVA 4 groups (based in previous publication about diet in Colombia). In 5.6% of the cases, it was not possible to break down the typical culinary preparations into their constituent ingredients (for example, "Cooked lasagna pasta," "Arepa de chocolo," "Tamal"). They were classified as minimally processed foods in the 'freshly prepared food' category, and those ready-to-eat fried, sweet or savory preparations were classified as processed foods (e.g., "buñuelos," "desserts," "fried empanadas").

### Assessing energy and nutrient intake

The energy content and CD-related nutrients (i.e., free sugars, total fats, saturated fats, trans fats, and sodium) in consumed foods were obtained from the Colombian Food Composition Table [19, 20] or the nutrient fact panels obtained from Colombian packaged foods. The nutritional information from packaged foods was collected in diverse supermarket chains in Colombia. Free sugars were estimated using the algorithm proposed by the nutrient profile model launched by the Pan American Health Organization [6]. The World Health Organization (WHO) nutrient intake goals were used to determine the prevalence of excessive intake of CD-related nutrients: $> = 10\%$ K.J. of total energy intake for free sugars, $> = 30\%$ K.J. of total energy intake for total fats, $> = 10\%$ K.J. of total energy intake for saturated fats, $> = 1\%$ K.J. of total energy intake for trans fats, and $> = 2000$ mg/8372 KJ for sodium [7, 22, 23].

## Sociodemographic characteristics

The sociodemographic characteristics available within ENDS included sex, age category (less than one y, 1-4y, 5-12y, 13-17y, 18-26y, 27-49y, 50-64y and pregnant woman), ethnicity (indigenous, Black/ Mulatto/Afro-Colombian and other ethnicities identified by self-reporting), area (urban/rural) and the wealth index makes it possible to compare the household economic conditions considering three aspects: Asset ownership, availability of utilities and home construction materials. The information was collected by a questionnaire sent to adult family members and households [20].

## Ethics statement

The research protocols have international and national endorsement from ethics committees in investigation and contemplated the return of the sampling results to the participants in ENSIN 2005 and 2015. The name of the Ethics Committees, the numbers or a statement that approval was granted by the named board(s), and a report that formal consent was obtained from the parent/guardian (including whether it was verbal or written) OR the reason consent is presented anonymously and confidentially in these technical reports [19, 20].

## Data analysis

First, we estimated the mean contribution to the total energy intake of each of the NOVA groups and subgroups, and a combination of groups in the consumption of culinary preparations combining energy intake from unprocessed or minimally processed foods and processed culinary ingredients, and of ready-to-eat food products (processed and UPF), to evaluate a proxy of consumption behavior by dietary patterns, in 2005 and 2015. Second, we estimated the mean content of CD-related nutrients (i.e., nutrients of which excessive intake according to the WHO thresholds has been linked to CD: energy density (KJ/g), free sugars (% of total energy intake), total fats (% of total energy intake), saturated fats (% of total energy intake), trans fats (% of total energy intake) and sodium (mg/8372 KJ)). It was done in the overall Colombian diet and two population diet fractions, one restricted to UPF items and the other limited to non-UPF items (unprocessed or minimally processed foods, processed culinary ingredients, and processed foods).

Third, we estimated the mean content of nutrients related to CD in the overall diet across quintiles of the dietary share of UPF. Linear regression models were used to test linear trends across quintiles. Crude- and sociodemographic-adjusted beta regression coefficients (β) were estimated to allow comparisons across variables with different measuring units. Fourth, we analyzed the association between the dietary share of ultra-processed foods (quintiles) and the frequency of dietary nutrient inadequacies using Poisson regression models, where the status of each individual regarding dietary inadequacy on each nutrient is the outcome variable (No: 0; Yes: 1) and quintiles of the dietary share of UPF (1–5) are the explanatory variable. Fifth, adjusted linear regression analyses were performed with sociodemographic indicators and all NOVA categories.

Finally, population attributable fractions (PAFs) were calculated to estimate the reduction in the prevalence of excessive intake of nutrients that, according to the WHO thresholds, has been linked to CD if the consumption of ultra-processed foods in Colombia was the one seen in the 20% lowest consumers of ultra-processed foods (the first quintile). The dietary share of ultra-processed foods among these consumers (n = 6820) represented 1.0% of total energy intake ranging from 0% to 4.5%. PAFs were calculated using the following equation:

$$PAF = \frac{P - Pq1}{P}$$

P is the prevalence of nutrient inadequacy in the population, and Pq1 is in the first quintile of ultra-processed food consumption. The analysis of the "Taylor series linearization variance approximation procedure" was used for variance estimation in all analyses to account for the complex sample design and the sample weights. Data were analyzed using the Stata 15. Excel was used for Figures.

## Results

In 2015, all foods and beverages' average daily energy intake was 8150 KJ. Natural or minimally processed foods accounted for 59.2% of total energy intake, processed culinary ingredients contributed 7.2%, and processed foods and UPF 14.4% and 19.2%, respectively (Table 1).

Table 1 also shows the contribution of different subcategories of products within the NOVA categories to energy intake. Within the natural or minimally processed foods category, those that contributed the most to energy intake were cereal grains, bananas, roots, and tubers (26%), followed by red meat with 9.2%. Freshly prepared foods (mainly prepared from unprocessed foods) accounted for 4.5% of total energy, fruits (not sources of vitamin A) contributed 3.3%, and dairy (milk/yogurt) and vegetables contributed 3.2% and 3.1%, respectively.

Among processed culinary ingredients, sugars were the most significant contributors to total energy (3.8%), followed by vegetable oils (3.1%). In the category of processed foods, the most significant contribution came from fried, salty, or sweet preparations (4.9%), followed by fresh bread and bakery products (3.1%) and cheeses (2.2%). Within the UPF, those that contributed the most to total energy intake were industrial bread (4.8%), sweet and salty packaged snacks (4.0%), sugar-sweetened beverages (3.7%), ice creams and commercial dairy drinks (2.3%), sausages and reconstituted meats (1.6%), were some of the other essential subcategories.

When evaluating the change in the consumption of foods and products by NOVA subgroups between 2005 and 2015, a decrease in the consumption of almost all food subgroups is observed in the group of natural or minimally processed foods, being more remarkable in the group of cereals, bananas, roots and tubers (-4.2%), followed by the group of homemade culinary preparations (-2.6%), the group of natural milk and yogurt (-2.3%) and legumes and legumes (-1.2%). On the contrary, there is an increase in the consumption of red meat (4.1%) and slightly of vegetables (1.5%). In the group of processed culinary ingredients, a decrease was observed in all the subgroups, being higher in sugars (-5.1%), followed by vegetable oils (-3.0%) and animal fat (-0.6%).

In the group of processed foods, an increase in consumption is observed in all subgroups, being higher in fried preparations (3.7%), followed by fresh bakery (1.4%) and slightly cheeses (0.3%). In the UPF group, an increase is observed in most of the subgroups, being higher in the subgroup of ice cream, industrial, commercial milk drinks (2.1%), followed by sweet and salty snacks (1.5%), sugar-sweetened beverages (1.2%), there was a slight decrease in confectionery (-0.8%).

Table 1 also shows the distribution of energy consumption in 2005 and 2015 in Colombia, according to the degree of food processing. In the groups of natural and minimally processed foods and processed culinary ingredients, a reduction in consumption is observed when comparing the years 2005 and 2015, of -4.1% and -8.6%, respectively. On the contrary, an increase in processed foods and UPF consumption has been observed, of 9.5% and 3.3%, respectively.

Making a combination of groups to evaluate a proxy of consumption behavior by dietary patterns, a decrease in the consumption of culinary preparations (-12.7%, combining energy intake from unprocessed or minimally processed foods and processed culinary ingredients) is observed between 2005 and 2015. On the contrary, when evaluating the trend in the

**Table 1. Mean and standard error consumption (Energy intake (%)) of food according to NOVA classification in Colombia between 2005 and 2015.**

| | 2015 y | 2005 y | Change in consumption |
|---|---|---|---|
| | \multicolumn Energy intake (%) | | |
| | mean (se) | mean (se) | Difference |
| **Group 1. Natural or minimally processed foods §** | **59.2 (0.2)** | **63.3 (0.3)\*** | **-4.1** |
| Cereals, roots y, tubers (includes flours) | 26 (0.2) | 30.2 (0.3)\* | -4.2 |
| Red meat | 9.2 (0.1) | 5.1 (0.1)\* | 4.1 |
| Traditional culinary preparations ⊥ | 4.5 (0.1) | 7.1 (0.2)\* | -2.6 |
| Fruits ‡ | 3.3 (0.1) | 3.6 (0.1) | -0.3 |
| Milk, yogurt (plain) | 3.2 (0.1) | 5.5 (0.1)\* | -2.3 |
| Vegetables | 3.1 (0.1) | 3.6 (0.0) | -0.5 |
| Eggs | 2.6 (0.1) | 2.5 (0.0) | 0.1 |
| Beans, legumes (includes flour) | 2.3 (0.1) | 3.5 (0.1)\* | -1.2 |
| Fish and shellfish | 0.7 (0.0) | 0.8 (0.0) | -0.1 |
| **Group 2. Processed culinary ingredients #** | **7.2 (0.1)** | **15.8 (0.2)\*** | **-8.6** |
| Sugar | 3.8 (0.1) | 8.9 (0.2) | -5.1 |
| Vegetable oils | 3.1 (0.1) | 6.1 (0.1) | -3.0 |
| Animal fat | 0.2 (0.0) | 0.8 (0.0) | -0.6 |
| **Group 3. Processed foods &** | **14.4 (0.2)** | **4.9 (0.1)\*** | **9.5** |
| Fried, salty, sweet processed foods | 4.9 (0.1) | 1.2 (0.1)\* | 3.7 |
| Bakery (fresh unpackaged) | 3.1 (0.1) | 1.7 (0.1)\* | 1.4 |
| Cheese | 2.2 (0.1) | 1.9 (0.1) | 0.3 |
| Meats (canned, smoked) | 0.1 (0.0) | 0.2 (0.0) | -0.1 |
| Canned fruits and vegetables | 0.03 (0.0) | 0.0 (0.0) | 0.0 |
| **Group 4. Ultra-processed food products ¢** | **19.2 (0.2)** | **15.9 (0.3)\*** | **3.3** |
| Industrialized breads | 4.8 (0.1) | 5.0 (0.1) | -0.2 |
| Snacks (sweet and salty) ∞ | 4.0 (0.1) | 2.5 (0.1)\* | 1.5 |
| Sugary drinks c | 3.7 (0.1) | 2.5 (0.1)\* | 1.2 |
| Ice Cream, Industrial Commercial Milk Drinks £ | 2.3 (0.1) | 0.2 (0.0)\* | 2.1 |
| Processed meats | 1.6 (0.0) | 1.3 (0.1) | 0.3 |
| Confectionery (chocolate, candies, sweets) | 0.7 (0.0) | 1.5 (0.1)\* | -0.8 |
| Ready-to-eat "junk food" preparations € | 0.6 (0.1) | 0.6 (0.0) | 0.0 |
| Industrial and commercial desserts | 0.4 (0.0) | 0.5 (0.0) | -0.1 |
| Industrial and commercial cereals | 0.4 (0.0) | 0.3 (0.0) | 0.1 |
| Energy drinks | 0.08 (0.0) | 0.0 (0.0) | 0.1 |

\* p<0.05 comparison between 2005 and 2015y

⊥ Includes pasta, sweet and savory foods that cannot be broken down into individual ingredients

‡Includes fruit pulp, coconut water

§ Includes cocoa, insect meat, coconut milk, soy milk, nuts, coffee, tea, tofu

# Includes spices, vinegar, yeast, vanilla extract, and plain gelatin.

& Includes canned fruits, canned vegetables, salted, sweetened, or in oil, roasted nuts or seeds, condensed milk, beer, and wine

∞ Includes chips, crackers, wafers, and cookies.

€ Including frozen foods, frozen pizza, soups, instant noodles

£ Includes custard, sweetened yogurts, milkshakes

¢ Includes spreads, margarine, broths, sauces, commercial baby foods, and distilled alcohol

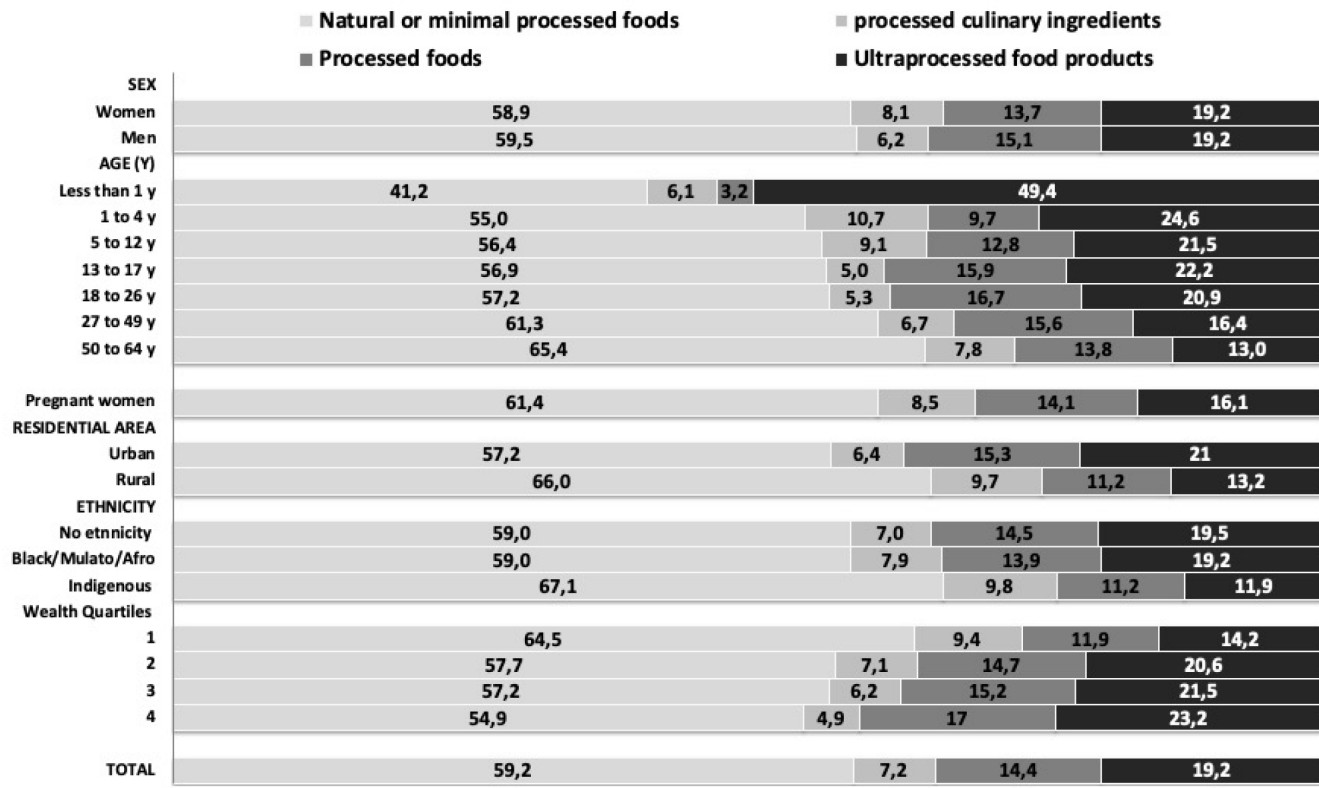

**Fig 1. Distribution of total energy intake according to NOVA classification and sociodemographic determinants in Colombia in 2015 y.**

consumption of ready-to-eat food products (processed and UPF), an increase (12.8%) in consumption is observed between the same years.

Fig 1 shows Colombia's share of UPF consumption by sociodemographic determinants in 2015. Those individuals that live in urban areas, with higher wealth index, children and adolescents, and without ethnicity or black/mulato/afro-descendant, according to the self-report, present the higher consumption of UPF in Colombia.

No significant differences were found between men and women in the consumption of ultra-processed foods. We observed significant differences by age, wealth index, area of residence, and ethnicity. The oldest participants had the highest consumption of natural or minimally processed foods. In contrast, the youngest participants (1–4 years old) had the highest intake of ultra-processed foods: more than double that of the participants aged over 50 years. The highest consumers of natural or minimally processed foods and the lowest consumers of ultra-processed foods were from the most lower wealth status and resided in rural areas.

The mean nutrient intake of the overall diet across quintiles of the dietary share of ultra-processed foods in ENSIN 2015 is presented in Table 2. From the lower to the upper quintile, there were increases in free sugars (from 8.1 to 15.4% of total energy intake), total fats (from 22.4 to 27.4% of total energy intake), saturated fats (from 8.1 to 11.1% of total energy intake), trans fats (from 0.21 to 0.60% of total energy intake), and sodium (from 1332 to 2169 mg/day). Trends of the increase of CD-promoting nutrients across quintiles of ultra-processed food consumption were statistically significant even after adjusting for potential sociodemographic confounders (P≤0.001).

**Table 2. The mean content of CD-related nutrients in the overall diet across quintiles of the dietary share of ultra-processed foods.** Colombian population aged one year or over (2015).

| Nutrient dietary content | Quintiles (Q) of the dietary share of ultra-processed foods (% of total energy) ‡ | | | | | Regression Coefficients [a] | |
|---|---|---|---|---|---|---|---|
| | Q1 | Q2 | Q3 | Q4 | Q5 | Crude | Adjusted [b] |
| Free sugar (% of total energy intake) | 8.1 | 10.2 | 11.1 | 12.7 | 15.4 | 1.71* | 1.64* |
| Total fat (% of total energy intake) | 22.4 | 24.7 | 24.8 | 25.9 | 27.4 | 1.12* | 0.90* |
| Saturated fat (% of total energy intake) | 8.1 | 9.0 | 9.3 | 10.2 | 11.1 | 0.83* | 0.62* |
| Trans fat (% of total energy intake) | 0.21 | 0.30 | 0.39 | 0.48 | 0.60 | 0.09* | 0.10* |
| Sodium (mg/day) | 1332 | 1712 | 1885 | 2079 | 2169 | 205.1* | 221.1* |

National Nutrition Examination Survey 2015 (n = 34600).

‡ Quintiles (Q) of the dietary share of ultra-processed foods (mean % of total energy (min and max)): Q1: (n = 6820): 1.0% (0.0, 4.5), Q2: (n = 6819): 8.7% (4.5, 12.5), Q3: n = 6819: 16.7 (12.5, 21.1), Q4: (n = 6819): 26.6 (21.1, 33.3), Q5: (n = 6819) 47.0 (33.3, 100).

[a]n Obtained from regressing dietary nutrient contents on quintiles of the dietary share of ultra-processed foods

[b] Adjusted for age, residential area, education level, and wealth index

*p<0.01 for linear trend across quintiles

Table 3 presents the prevalence of nutrient intake inadequacies for the overall population and across quintiles of the dietary share of ultra-processed foods. The higher prevalence of CD-related nutrients excess consumption was found for sodium (67.6%), followed by free sugars (48.6%), saturated fats (40.8%), total fats (26.7%), and trans fats (9.7%). The prevalence of CD-related nutrients excess consumption significantly increased with ultra-processed foods, even after adjusting for potential sociodemographic confounders (P≤0.001). Thus, compared with individuals in the lowest quintile, individuals in the highest quintile of ultra-processed food consumption were significantly more likely to have diets exceeding the dietary goals for trans fats (5.7 times), free sugars (2.7 times), sodium (2.3 times), saturated fats (2.1 times) and total fats (1.9 times).

**Table 3. Prevalence of diets with excessive content of CD-related nutrients [δ] in the whole population and across quintiles of the dietary share of ultra-processed foods.** Colombian population aged one year or over (2015).

| Nutrient | Whole population | Quintiles (Q) of the dietary share of ultra-processed foods (% of total energy) ‡ | | | | | P.R [a] | |
|---|---|---|---|---|---|---|---|---|
| | | Q1 | Q2 | Q3 | Q4 | Q5 | Crude | Adjusted ♮ |
| | | Individuals who did not meet the recommendation (%) * | | | | | | |
| Free sugars | 48.6 | 27.2 | 39.5 | 48.8 | 58.2 | 72.8 | 1.45* | 1.25* |
| Total Fat | 26.7 | 19.4 | 24.8 | 24.9 | 29.4 | 36.4 | 1.15* | 1.08* |
| Saturated fats | 40.8 | 28.1 | 32.8 | 38.0 | 49.3 | 58.8 | 1.21* | 1.16* |
| Trans Fat | 9.7 | 3.9 | 5.1 | 5.8 | 13.1 | 22.3 | 1.63* | 1.71* |
| Sodium | 67.6 | 36.2 | 62.6 | 75.1 | 82.3 | 83.7 | 1.18* | 1.15* |

National Nutrition Examination Survey 2015 (n = 34600).

P.R. = Prevalence ratios estimated using Poisson regression

δ See methods for cut-offs used to define inadequate dietary nutrient contents.

‡ Quintiles (Q) of the dietary share of ultra-processed foods (mean % of total energy (min and max)): Q1: (n = 6820): 1.0% (0.0, 4.5), Q2: (n = 6819): 8.7% (4.5, 12.5), Q3: n = 6819: 16.7 (12.5, 21.1), Q4: (n = 6819): 26.6 (21.1, 33.3), Q5: (n = 6819) 47.0 (33.3, 100).

[a] Obtained from a logistic regressing model of the inadequacy of dietary nutrient intake on quintiles of the dietary share of ultra-processed foods

♮ Adjusted for age, residential area, education level, and wealth index

*p < 0.001, for linear trend across quintiles of ultra-processed food consumption

*p<0.01 for linear trend across quintiles

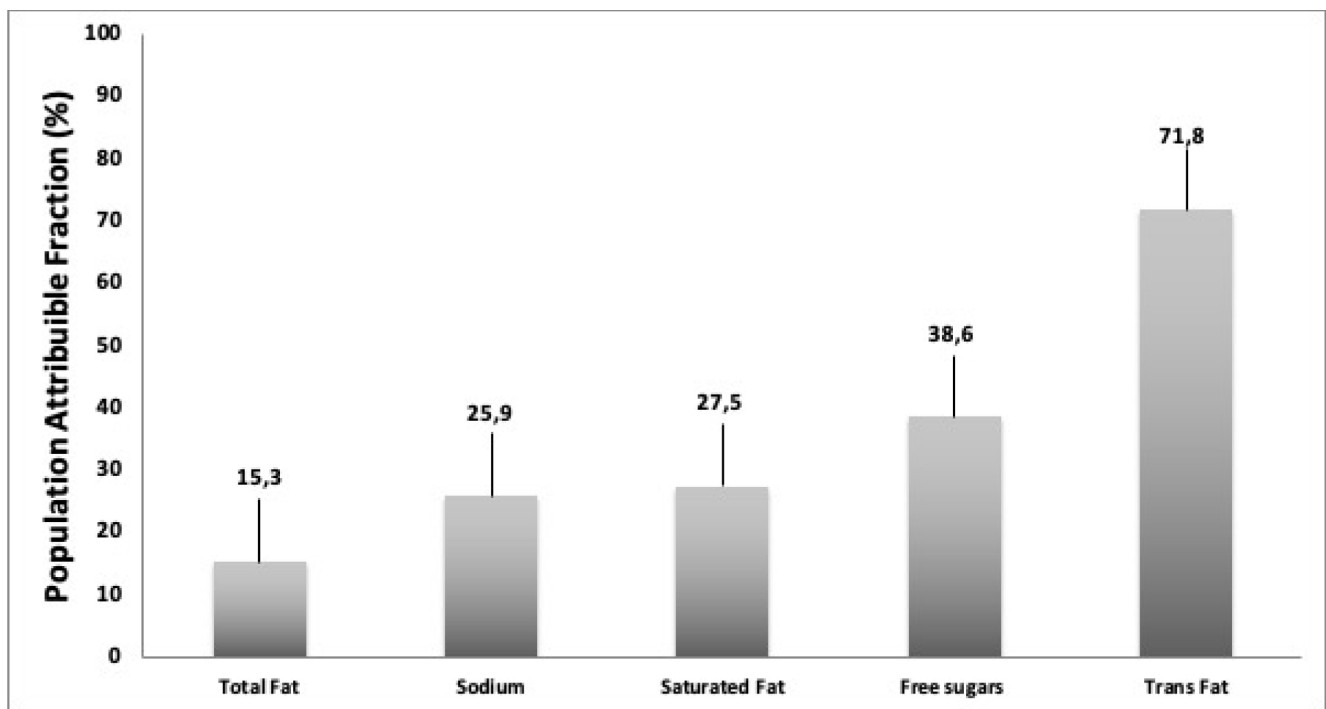

**Fig 2. Population attribuible fraction due to the high consumption of ultra-processed foods (higher than the observed in the first quintile) of inadequate NCD-related nutrients dietary content.** Colombian population aged one years or over (2015).

Fig 2 presents estimates of the fractions of the populations that have an inadequate intake of CD-related nutrients attributable to high consumption of ultra-processed foods in Colombia (higher than the one found among the 20% lowest consumers of UPF, who consume 1.0% of the total energy intake on average, ranging from 0 to 4.5%). Three-fourths of the population's excessive intake of trans fats; around one-third of the population's excessive intake of free sugars and saturated fats; nearly 26% population's excessive intake of sodium; and 15% population's excessive intake of total fats; are due to the referred excessive intake of UPF.

## Discussion

In Colombia, there is an increase in the consumption of UPF. In 2015, no significant differences were found in the consumption of UPF between men and women. But, significant differences by age, wealth index, area of residence, and ethnicity. A significant positive association was found between the dietary share of UPF and the intake of CD-related nutrients such as content of free sugars, total fats, saturated fats, trans-fats, and sodium. The prevalence of excessive intake of all these nutrients (WHO recommendations) increased across quintiles of the dietary share of UPF. In concordance, with the reduction of UPF consumption to the level seen among the 20% lowest consumers, the prevalence of excessive nutrient intake would be reduced in 72% for trans fats; in 39% for free sugar, 27.5% in saturated fats, in 26% for sodium and 15% for total fat.

These observed trends of increased consumption of UPF in Colombia are consistent with the data described by PAHO on the increase in the sale of UPF in recent decades in the country and the region [24]. It is essential to highlight that parallel to the risk associated with the increase in the presence and consumption of these unhealthy UPF in the country, a decrease in the consumption of natural or minimally processed foods and culinary preparations is

observed. Both phenomena (increase in the consumption of UPF and decrease in the consumption of culinary preparations based on natural foods) generate a deterioration in the quality of the diet due to an increase in the content of critical nutrients related to the presence of obesity and chronic diseases (i.e., free sugars, total, saturated and trans fats, and sodium) and the reduction of dietary diversity of natural foods [25], and essential nutrients for health like micronutrients, fiber and protein [11].

In Colombia, in 2015, rural residents and older adults are likely to have more traditional cooking and eating practices and more stable dietary patterns. These groups may also be more protected from marketing practices that appeal to the younger generation and more advantaged people with less stable dietary habits and, therefore, more likely to try these products. These findings are in line with the ENSIN 2005 results [10].

In light of the substantial amount of recent research accentuating the risks of UPFs [5], the results presented above reflect an alarming situation which has been associated with the corporate food regime [26]. According to Ordoñez, this regime emerged in the 1990s in Colombia [17, 27], presenting the following characteristics: (i) institutional frameworks and functional regulatory frameworks for the industrial agri-food system; (ii) privatization of seeds and the restriction of their conservation and free circulation; (iii) intensive use of chemical poisons and fertilizers in agriculture; (iv) corporate concentration of its production, import and sale; (v) land grabbing by national and foreign businessmen in large regions of the country such as the "Altillanura"; (vi) deepening of agro-industrial bets and the extension of monocultures; (vii) expansion and increase of food distribution chains in large supermarkets and express formats; (viii) proliferation of UPF sales (of national and international origin), and fast-food chains in the different food environments [24]; (viii) increase obesity, chronic diseases and standardization of people in a situation of malnutrition; and (ix) he precise definition of a vital business bloc in the agricultural and food sectors. Additionally, this trend in the increase of UPF may be due to the different marketing and advertising techniques used by the food industry, which is based on the management of emotions, for the sale of their products, such as selling: fun/ happiness, fantasy and imagination, palatability, drawings or animated characters, music/jingles and messages aimed at children [28].

According to Mialon et al. [16, 29], proximity between the industry, government, and the media is particularly evident and remains unquestioned in Colombia. The influence of vulnerable populations in communities and the feeling of insecurity by public health advocates is also problematic. The Corporative Political Activity of the food industry has the potential to weaken and delay efforts to develop and implement public health policies that could improve the healthiness of food environments. Mechanisms to prevent and manage the influence of the food industry must be developed in the country [16, 29–33].

The results of this study from 2015 support the hypothesis that strategies to reduce the consumption of UPF are a potentially effective way to achieve the nutritional goals of the WHO for the prevention of diet-related CD. We observed that a reduction of UPF consumption to the level seen among the 20% lowest consumers significantly reduced the prevalence of excessive CD-related nutrients intake. In concordance, Martinez and Cols also found that inadequate intakes of these nutrients would be reduced considerably in all eight countries studied (Brazil, Chile, Canada, Mexico, Australia, UK, US, and Colombia with data from 2005) if ultra-processed food consumption were decreased to levels observed among the lowest quintile of UPF consumption in each country [9].

The results of this study highlight the need for regulations of UPFs, including:

(i) implementation of law 2120 approved by the Colombian parliament to improve the food environments to prevent CD [34].

(ii) disseminate health promotion tools, such as plant-based dietary guidelines that consider the extent and purpose of processing, such as those adopted in Brazil [35] and Uruguay [36], where have these recommendations: Make natural or minimally processed foods the basis of your diet, use oils, fats, salt, and sugar in small amounts when seasoning and cooking natural or minimally processed foods and to create culinary preparations, limit consumption of processed foods, and avoid consumption of ultra-processed foods.

(iii) implementing the warning labeling to encourage healthier options at points of purchase, such as the octagonal warning labels adopted in several Latin American countries, as recommended by PAHO [37–39]

(iv) counteracting the different strategies of political activity by the corporate food regime that has been established in Colombia [16, 29]

(v) Adopt healthy taxes on UPF products high in CD-promoting nutrients and the presence of sweeteners

(vi) apply the PAHO nutrient profile model in food policies to regulated de UPF, which has demonstrated in nine countries (including Colombia) that those individuals that consume UPF or processed products with excess content of nutrients related to a CD according to PAHO presented two to four times more probability to have a diet associated with a risk of CD [6, 40]

(vii) regulate UPF advertising for children and adolescents complying with the WHO set of Recommendations on the Marketing of Food and Non-Alcoholic Beverages to Children.

(viii) improve food systems so that they are healthy and sustainable, based on clean production, avoiding the UPF in social programs, support and subsidies for local producers of natural foods, short marketing circuits, and supported by food and nutrition education strategies.

This study has its limitations. All dietary information available within ENSIN was of 24-hour dietary recall. Beyond recall bias, participants may choose to retain information about certain food products considered socially undesirable in terms of product categories and quantities consumed. This single point of dietary information may not capture the participants' usual diet and therefore be less representative of their intake. Nonetheless, the probabilistic nature of the sample studied, the national representativeness of the Colombian population using national surveys with available 24-hour dietary recall, and the standardization of dietary data collection are some of the strengths of this study.

In addition, it also represents a strength, the fact that the study provides intake reference levels using the ENSIN 2005 as a baseline and compares with the most recent dietary evaluations in the country (ENSIN 2015), capturing trends in the consumption of UPF in Colombia. Additionally, this work considers the nutrients related to chronic diseases that are the focus of the recent food policy on Front of package labeling and healthy taxes on UPF in the country. Finally, the analysis of PAFs of unhealthy nutrient intake in Colombia in 2015 due to ultra-processed food consumption could be a reference as a baseline for follow-up in the political to reduce ultra-processed foods in Colombia. These results suggest an installation of the corporate food regime in Colombia, displacing the ancestral traditional food system related to a diet based on natural foods and typical culinary preparations associated with a healthy and sustainable diet. These data highlight the need to improve the current health promotion and prevention policies reducing the sale of UPF to achieve the nutritional goals of the WHO for the prevention of diet-related CD.

## Conclusions

This study shows an increasing trend in the consumption of UPF by the population of Colombia from 2005 to 2015 and a decrease in the consumption of natural and minimally processed

foods and culinary preparations (except red meat). Children and adolescents were found to have a higher consumption of UPF, which is probably due to the fact that they are more easily hooked up by marketing practices of UPF and, therefore, are more vulnerable to their deleterious effects on diet, increasing the consumption of CD-related nutrients. Adopting low-UPF diets, following the 20% lowest consumers of UPF ($\leq$1.0% (0–4.5%) of the total energy from UPF), would allow the prevalence of unhealthy diets among the Colombian population to reduce significantly. Strategies to reduce the consumption of UPF are a potentially effective way to achieve the nutritional goals of the WHO for the prevention of diet-related CD.

## Supporting information

**S1 Appendix.**
(DOCX)

**S1 Fig. Food consumption, according to NOVA classification and sociodemographics factors in Colombia.**
(XLSX)

## Acknowledgments

Thanks to Colombia's Ministry of Health and social protection for providing the database.

## Author Contributions

**Conceptualization:** Gustavo Cediel.

**Formal analysis:** Gustavo Cediel.

**Funding acquisition:** Gustavo Cediel.

**Methodology:** Gustavo Cediel, Elisa María Cadena, Pamela Vallejo, Diego Gaitán, Fabio Da Silva Gomes.

**Resources:** Pamela Vallejo.

**Supervision:** Pamela Vallejo.

**Validation:** Elisa María Cadena, Fabio Da Silva Gomes.

**Visualization:** Diego Gaitán, Fabio Da Silva Gomes.

**Writing – original draft:** Gustavo Cediel, Pamela Vallejo, Diego Gaitán, Fabio Da Silva Gomes.

**Writing – review & editing:** Gustavo Cediel, Elisa María Cadena, Pamela Vallejo, Diego Gaitán, Fabio Da Silva Gomes.

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
