## [Decision Letter · Decision Letter 0]

10 May 2023

PGPH-D-23-00294

The increasing trend in the consumption of ultra-processed food products is associated with a diet related to chronic diseases in Colombia. Evidence from National Nutrition Surveys 2005 and 2015.

Dear Dr. Cediel,

Thank you for submitting your manuscript to PLOS Global Public Health. After careful consideration, we feel that it has merit but does not fully meet PLOS Global Public Health’s publication criteria as it currently stands. Therefore, we invite you to submit a revised version of the manuscript that addresses the points raised during the review process.

We look forward to receiving your revised manuscript.

Kind regards,

Leonor Guariguata, MPH, PhD

Academic Editor

Journal Requirements:

1. We noticed you have some minor occurrence of overlapping text with the following previous publication(s), which needs to be addressed:

- 10.1017/S1368980019004737

- https://doi.org/10.11606/s1518-8787.2020054001176

In your revision ensure you cite all your sources (including your own works), and quote or rephrase any duplicated text outside the methods section. Further consideration is dependent on these concerns being addressed.

2. Please ensure that Funding Information and Financial Disclosure Statement are matched.

3. In the Funding Information you indicated that no funding was received. Please revise the Funding Information field to reflect funding received.

4. Please provide separate figure files in .tif or .eps format only and remove any figures embedded in your manuscript file. Please also ensure all files are under our size limit of 10MB.

5. We have noticed that you have uploaded Supporting Information files, but you have not included a list of legends. Please add a full list of legends for your Supporting Information files after the references list. 

Additional Editor Comments (if provided):

Reviewers' comments:

Reviewer's Responses to Questions

**Comments to the Author**

1. Does this manuscript meet PLOS Global Public Health’s publication criteria? Is the manuscript technically sound, and do the data support the conclusions? The manuscript must describe methodologically and ethically rigorous research with conclusions that are appropriately drawn based on the data presented.

Reviewer #1: Yes

Reviewer #2: Yes

2. Has the statistical analysis been performed appropriately and rigorously?

Reviewer #1: Yes

Reviewer #2: I don't know

3. Have the authors made all data underlying the findings in their manuscript fully available (please refer to the Data Availability Statement at the start of the manuscript PDF file)?

Reviewer #1: Yes

Reviewer #2: No

4. Is the manuscript presented in an intelligible fashion and written in standard English?

Reviewer #1: No

Reviewer #2: No

5. Review Comments to the Author

Reviewer #1: I would like to thank the Editor and authors for the opportunity to review this manuscript. The topic is very relevant due to the increased concern of the role of ultra-processed foods in the nutrition transition underway, particularly in low and middle-income countries. The authors analysed data from a nationally representative sample of Colombians, providing first time evidence of trends using individual-level consumption data. Some suggestions to improve the quality of the manuscript are presented below:

General comments:

- The manuscript needs proofreading throughout (e.g., line 148).

- It is unclear how objectives ii and iii add to previous evidence using 2005 data. The originality and relevance of new analysis should be explained further in the introduction and discussion; e.g., the rising in UPF consumption could explain changes in prevalence of inadequate intake. For that end, instead of presenting aims ii and iii only with 2015 that, would be interesting to see changes in prevalence between 2005-15 attributable to increased intake of UPF. For instance, following this methodology: doi:10.3389/ijph.2022.1604103

- Table 1 requires t-test of mean differences to state that there are significant temporal variance.

- Figure 1 is a good visual, but would be better if accompanied by a table (could be supplementary material) presenting coefficients of associations (and respective 95%CI or p-values) of UPF consumption among different strata of the population. Similarly, would be good to present the variation in UPF consumption between 2005-2015 in different sociodemographic groups.

- Although methods and discussion mention the use of standardized coefficients, these do not seem to be presented in Table 2.

- Could you please provide as supplementary material or in the main manuscript beta coefficients of the association of UPF consumption continuous and CD-nutrient intakes of Table 2? This is useful because quintile 5 has a large variability in UPF intake (33 to 100%).

- Figure 2 methods, results and discussion: if the level represented is the one among the 20% lowest consumers, UPF intake was then set to the upper bound (4.5%) instead of 1%?

- Discussion is lacking explanation why more advantaged people have higher intakes of UPFs in Colombia.

Minor comments:

Abstract:

- Methods: replace “degree and extent” with “extent and purpose”

- Results: Please provide p-value considered significative

- Conclusions: Missed comments related to the time trends and implications of UPF to the decreasing in the quality of diets in Colombia.

Line 138: sociodemographics: missing rural/urban residence and description of pregnant women presented in Fig 1. It is unclear why pregnant women were included in Fig 1.

Table 2: Sodium is mg/day or density?

Table 3: ** p<0.05 can be removed

Line 78: missing “respectively” in the end of the sentence

Line 80: UPF consumption instead of use?

Line 82: Would be good to provide evidence on the prevalence of chronic diseases (e.g., obesity) in Colombia, particularly if there was growth.

Line 115: extent and purpose? Keep it consistent throughout.

Line 118: subgroups within the NOVA 4 groups?

Line 103: age >1y instead?

Thank you very much again for the opportunity to review this manuscript.

Reviewer #2: General

First off I would like to congratulate the authors on the interesting study they conducted. Nonetheless, the manuscript would benefit from being thoroughly revised as to improve the strength and readability.

• At times the language is unclear, making it difficult to follow. I would advise the authors to work with a native English speaker or English speaking copyeditor to improve the readability.

• The term ‘Chronic diseases (CD)-promoting nutrients’ is used throughout the text to refer to sugar, fat (total, saturated and trans) and sodium. This seems a strong statement to me. There is a strong link between diets and chronic diseases, but is there sufficient evidence to support a direct link with nutrients?

• The introduction needs to be strengthened. Currently it provides numerical information, but does not clearly lead up to the objectives. For example, in the discussion the corporate food regime and corporate political activity are brought up. It could be interesting to already add this in the introduction leading up to the objectives of the study.

• Methods needs to be more clear and to the point:

o Make a more clear distinction between what you as researchers/authors did and what was obtained from existing surveys. For example, ENSIN data were 24-hour recall data, but when in the text describing these data it sometimes seems these were collected by the researchers

o When reading the results it says ‘making a combination of groups with the aim of evaluating a proxy of consumption behavior by dietary patterns…’. It would be beneficial to already describe what was done, why and how in the methods section.

• The current results section often describes in rather lengthy detail what is shown in the table. In my opinion this section could benefit from in text solely writing the important results and referring to the tables for the details.

• Although there is a lot of very interesting information within the current discussion, it feels a little all over the place. I would recommend to focus on discussion points directly stemming from the results from the study and placed within the context introduced at the beginning (within the introduction).

Below I have included some thoughts that came to mind when reading the different sections, which might help clarify some of the issues mentioned above.

Abstract

• Tine layout remark: I would start each section (e.g. objectives, methods,…) on a new line to create a better overview.

• In the beginning, maybe add at least one introductory sentence. Just listing the objectives feels like a strange beginning.

• Line 36-37: There is a strong link between diets and chronic diseases, but is there sufficient evidence to support the claim ‘…CD-promoting nutrients such as…’. In my opinion this might be too strong a statement.

• Line 39-40: ‘excessive intake of all CD-promoting nutrients (according to WHO recommendations)’ - CD-promoting nutrients according to WHO or nutrients above the thresholds defined by WHO? Not 100% clear to me.

Introduction

• Apart from the first paragraph, the current introduction provides a lot of interesting numerical information, but does not clearly lead up to the objectives. The introduction could be made stronger by making it more clear why this study is so important within the current setting and context.

• Line 65-66: ‘…these products in Latin America, with a 26% increase between the year 2000 and 2013, being 25%, the increase in Colombia for the same period.’ Not completely clear. Do you mean that the trend in Colombia was similar with a 25% increase over the same time period? If so, split up into two sentences.

• Line 67: remove from (‘national level from in 2005-> national level in 2005)

• Line 68: change to ‘came from UPF’ – was in the past (2005).

• Line 66-70. Sentence is rather long. Maybe split up into two sentences to make it more clear? E.g. ‘Additionally, analysis of food consumption at the national level in 2005 showed that 15.9% of energy consumed by the population of Colombia came from UPF. The highest consumption was observed among adolescents with a contribution of 20.3% to the total energy consumed per day.’.

• Line 72: Move ‘in Colombia to after ‘all types of UPF’.

• Line 73-74: Frozen products, canned products, preserved products and dairy products aren’t necessarily ultra-processed.

• Line 76-78: Is this overarching for Brazil, Chile and Mexico? Or respectively for each country?

• Line 76: comma before such as.

• Line 77: comma after Mexico.

• Line 78: of-> with

• Line 82-85: Apart from the first sentence, this paragraph does not clearly lead up to the objectives.

o Nothing has been mentioned before on foreign investment within the introduction.

o Nothing has been mentioned on overweight and obesity in Colombia within the introduction.

o Apart from the first paragraph, the introduction does not reflect well the issue of UPF.

• In the discussion (line 368-372) the study of Martinez and cols is mentioned which was conducted across 8 countries, including Colombia, and had similar findings. This immediately made me wonder about the added value of this study. There definitely is an added value as you had data from 2005 and 2015 and (from the current text), seem to have gone more into depth, but this needs to be made more clear in the introduction leading up to objectives.

Methods

• Line 94-95: remove ‘y’ after the dates in brackets.

• Line 96-97: between the brackets change to ‘ENSIN 2005 and 2015’.

• Line 102-111: From this paragraph it sounds as if the authors did this, but in the previous paragraph it says that the data from ENSIN were used for this. Assuming that this is about how the ENSIN data were collected, reformulate the paragraph to make this clear.

• Line 118: 33 subgroups? Only explained the NOVA classification above (with 4 groups), so please explain what is meant here.

• Line 118-123: Not clear what is meant. Please clarify.

• Line 126: energy content.

• Line 126: what type of sugars? Added, total,…?

• Line 132: ‘were used for evaluating the prevalence of non-recommended intake of different key nutrients’ -> Were used to determine the prevalence of excessive intake of nutrients of concern? Non recommended intake can also be too little, while I assume the focus here is on too much.

• Line 138-145: About extra data collected by authors or about the data available within ENDS?

• Line 147-154: Improve this paragraph. Currently there is quite some rather vague information there.

• Line 159: CD-promoting nutrients seems like a too strong choice of words. Maybe rather something in the lines of ‘nutrients of which excessive intake according to the WHO thresholds has been linked to CD?

• Line 165: Minor questions, but why did you chose for quintiles?

• Line 176-177: ‘reduction in the prevalence of nutrient inadequacies’ -> Inadequate sounds like people aren’t getting enough, but I suppose that you want to say the opposite.

Results

• The text currently repeats a lot what is already shown in Table 1 (e.g. line 200-232). In the text I would suggest to focus on the overarching findings that are interesting and reference to Table 1 for the details.

• Table 1: the ‘se’ in brackets after the mean, does this refer to the standard deviation? If so, please clarify in the title.

• Line 200-201: ‘Table 1 also shows the contribution of different subcategories of products within the NOVA categories to energy and nutrients intake’ – Within the current table I only see the contribution to energy, not nutrients.

• Line 237: In text says -4.2% while in the table it says -4.1%. Check which one is correct and adjust.

• Line 241: Making a combination of groups? Please add some information on this in the methods section

• Figure 1: Currently the numbers are rather small. Maybe increase the overall figure?

Discussion

• The current discussion doesn’t always flow naturally from the results that were really found during this study. E.g. on line 319 it suddenly mentions ‘unsustainable’ which is less relevant for this article and these findings.

• Line 334: ‘in analysis from 2005’. Which analysis? Please provide some more information.

• Line 336-349: In the introduction the corporate food regime was hardly mentioned or introduced. This makes it very strange to then here have such a long paragraph about this.

• Line 355-362: Same as for the above. The Corporative Political Activity (CPA) was not yet touched upon in the introduction, which makes it a little strange to have this here. Both corporate food regimes and CPA could however be very interesting to include in the introduction if you feel this is an important reason for doing this study and context for the results.

• Line 376: Law 2120 – Please provide some context. Most readers (like me) won’t know this law.

• Line 378-379: Shortly explain the examples of Brazil and Uruguay

• Line 378: Apply the PAHO nutrient profile model for what? Restricting marketing? Accessibility of products?

• Line 394-396: Although I completely agree, I don’t see how this recommendation flows from the results of this study.

• Line 398: Was collected? This gives the impression that this was done within this study, but earlier I had understood that this were the ENSIN data. Please be specific about this.

Conclusions

• The conclusion could be a little more focused on the results itself and the most important conclusions. Currently it still feels a little too much like a continuation of the discussion.

6. PLOS authors have the option to publish the peer review history of their article (what does this mean?). If published, this will include your full peer review and any attached files.

**Do you want your identity to be public for this peer review?** For information about this choice, including consent withdrawal, please see our Privacy Policy.

Reviewer #1: No

Reviewer #2: No

---

## [Decision Letter · Decision Letter 1]

25 Oct 2023

PGPH-D-23-00294R1

The increasing trend in the consumption of ultra-processed food products is associated with a diet related to chronic diseases in Colombia. Evidence from National Nutrition Surveys 2005 and 2015.

Dear Dr. Cediel,

Thank you for submitting your manuscript to PLOS Global Public Health. After careful consideration, we feel that it has merit but does not fully meet PLOS Global Public Health’s publication criteria as it currently stands. Therefore, we invite you to submit a revised version of the manuscript that addresses the points raised during the review process.

Dear authors,

There are just some minor edits to include and then we can move forward with the manuscript. Thank you for your work.

Kind regards,

Leonor

We look forward to receiving your revised manuscript.

Kind regards,

Leonor Guariguata, MPH, PhD

Academic Editor

Journal Requirements:

Additional Editor Comments (if provided):

Reviewers' comments:

Reviewer's Responses to Questions

**Comments to the Author**

1. If the authors have adequately addressed your comments raised in a previous round of review and you feel that this manuscript is now acceptable for publication, you may indicate that here to bypass the “Comments to the Author” section, enter your conflict of interest statement in the “Confidential to Editor” section, and submit your "Accept" recommendation.

Reviewer #1: All comments have been addressed

Reviewer #2: (No Response)

2. Does this manuscript meet PLOS Global Public Health’s publication criteria? Is the manuscript technically sound, and do the data support the conclusions? The manuscript must describe methodologically and ethically rigorous research with conclusions that are appropriately drawn based on the data presented.

Reviewer #1: Yes

Reviewer #2: Yes

3. Has the statistical analysis been performed appropriately and rigorously?

Reviewer #1: Yes

Reviewer #2: Yes

4. Have the authors made all data underlying the findings in their manuscript fully available (please refer to the Data Availability Statement at the start of the manuscript PDF file)?

Reviewer #1: Yes

Reviewer #2: Yes

5. Is the manuscript presented in an intelligible fashion and written in standard English?

Reviewer #1: Yes

Reviewer #2: No

6. Review Comments to the Author

Reviewer #1: (No Response)

Reviewer #2: General

First off, I would like to congratulate the authors on the major improvements made to the manuscript, particularly the methods and results section. Nonetheless, some suggestions remain to improve the overall quality of the manuscript. Overall I would suggest to:

• Have someone thoroughly check the English, particularly the construction of some sentences could be improved (some examples provided below).

• Change the term ‘Chronic diseases (CD)-promoting nutrients’ to ‘Nutrients of concern’ which (in my experience) is a more accepted term. In case it is chosen to stick with CD-promoting nutrients or NCD-related nutrients (as in Table 2 and Table 3), make sure to use the same terminology throughout the manuscript. For example, in the discussion also the term critical nutrients is introduced.

• Improve the relation between the introduction and the discussion. Both currently still feel a little all over the place, especially the discussion. I would recommend to choose the core topics you want to discuss in both sections and double check that no new topics are discussed in detail in the discussion section which have not been introduced in the introduction.

Abstract

• The abstract would benefit from some reorganisation. Currently it still lacks a little information in the background section while it goes very much into detail in the results section. I would recommend to really focus on the key findings.

• In the conclusion of the abstract it is stated that ‘the increasing trend in the consumption of UPF is associated with diet-related CD’. This isn’t a conclusion that can be made solely on these results as this study did not look at CD, solely at nutrients of concern/ CD-promoting nutrients. It would be good to formulate a more accurate and to the point conclusion.

Introduction

• Line 66-67: ‘The Pan American Health Organization (PAHO) reported a growing trend in the sales of these products in Latin America’ -> Change ‘these products’ to ‘UPFs’. Remove the comma at the end of the sentence (after 2013).

• Line 67-68: ‘In Colombia, it was similar, being 25%, the gain for the same period’ -> Change to something in the lines of ‘In Colombia a similar increase of 25% over the same period was observed’.

• Line 70-71: ‘The highest consumption was observed among adolescents, contributing 20.3% to the total daily energy consumed’ -> Change to something in the lines of ‘The highest consumption was observed among adolescents, with UPF contributing 20.3% to the total daily energy intake’.

• Line 80-81: The last sentence linking to the corporative food regime appears rather suddenly. Maybe add some literature about the link between the corporate food regime and the increased sales of UPFs. The same goes for the next sentence (line 84-87) which could be in the same paragraph as the previous one. As this aspect is also thoroughly discussed later on it would make sense to allocate one paragraph on the food industry in the introduction.

Methods

• Overall I would recommend to make it clear which explanation is about the work that the authors/researchers did themselves and which parts are rather an explanation about existing databases (ENDS and ENSIN).

• Line 98: Idem as for ENSIN between brackets, change ‘(ENDS, 2005, 2015)’ to ‘(ENDS 2005 and 2015)’.

• Line 103-112: From how the paragraph is currently formulated it still seems as if the authors/researchers collected the dietary data, while these data seem to be obtained from ENSIN. Still assuming that this is about how the ENSIN data were collected, I suggest to make this more clear in this paragraph (e.g. start the paragraph with ‘The dietary data within ENSIN were obtained using….’).

• Line 119: Currently it remains unclear how these 33 subgroups were decided upon. Is it based on a certain classification? Based on how the ENSIN database is organised?

• Line 130-131: ‘The nutritional information from packaged foods was collected in significant supermarket chains in Colombia.’ -> Collected by who is in charge of the Colombian food composition table or by the authors/researchers? Moreover, what is meant by significant supermarket chains? Significant because of their market share in general in the country? Or because of there presence in bigger cities? How were the supermarkets selected?

• Line 136: Add fats after trans (~trans fats).

• Line 140: Do you mean the sociodemographic information available within ENDS? If sociodemographic information was already available in ENSIN, for what exactly was the ENDS information used?

• Line 146-147: ‘The collected data was classified into one of four levels.’ -> Four levels of what?

• Line 162: Remove bracket after ingredients.

Results

• Figures: The current figures (Figure 1 and Figure 2) as attached to the manuscript need to be sharpened to make them readable.

• Tables: Just a minor detail, but it would be nicer to have the legend in order of appearance of the symbols in the table.

• Line 249-252: Reorganise this sentence to something like: Making a combination of groups to evaluate a proxy of consumption behavior by dietary patterns, a decrease in the consumption of culinary preparations (-12.7%, combining energy intake from unprocessed or minimally processed foods and processed culinary ingredients) is observed between 2005 and 2015.

• Line 261-268: In the discussion it is mentioned ‘that people aged below 19 years, in the higher wealth index quintiles, living in urban areas, and without ethnicity or black/mulato/afro-descendant, according to the self-report, were the primary consumers of UPF’. In the current manuscript this result doesn’t seem to be reported in the result section yet. Maybe something about this could be added to this paragraph?

• Line 267: Most inferior wealth status sounds a little condescending. I would suggest to formulate this in a more neutral way (potentially something like ‘from the lower wealth index quintiles’ or ‘had a lower wealth status’).

• Line 292: As earlier the abbreviation PAF was introduced for population attributable fractions, maybe stick to using this same abbreviation?

• Line 292-293: Change ‘inadequate intake’ to ‘excessive intake’?

Discussion

• Overall I would recommend to discuss the results more thoroughly. Currently there are a lot of longer lists with characteristics and suggestions for regulations in comparison to the extent in which the results are discussed.

• The first paragraph of the discussion provides a very nice summary!

• Line 309-311: Change to ‘would be reduced to’. Moreover I would suggest to use the same expression for all the nutrients (fractions or percentages).

• Line 320: Here a new terminology is introduced ~ ‘critical nutrients’ instead of ‘CD-promoting nutrients’. Stick to the same terminology throughout the manuscript.

• Line 325-332: This is a result from the study, so important to add this to the result section (for example to paragraph line 261-268; see suggestion above).

• Line 331-332: Maybe simplify this last sentence to something like ‘These findings are in line with the ENSIN 2005 results (ref)’.

• Line 334: Remove ‘alarming’, or, as I do agree that it is kind of alarming, explain better to the reader why this situation is alarming. This could be done, for example, by adding some more literature on the risks of UPFs (~In light of the substantial amount of recent research accentuating the risks of UPFs (references), the results presented above reflect an alarming situation which has been associated with the corporate food regime…’.

• Line 335: Since-> in (emerged in the 1990s).

• Line 345: Can remove ‘in the population’.

• Line 345-346: Minor detail - before the different characteristics didn’t start with a capital letter and from (viii) onwards they do. I would recommend to be consistent.

• Line 354: Remove mainly.

• Line 356-359: This is very true and highly important, but within the discussion I would stick to discussing issues related directly to the findings of this study.

• Line 365-369: As Martinez and Cols found similar results, including for Colombia, it might be good to add in the introduction what the added value is of this study compared to theirs.

• Line 367: The questions that immediately came to mind was, which eight countries. Maybe add between brackets?

• Line 371: Maybe slightly reformulate this sentence? For example ‘The results of this study highlight the need for regulations of UPFs…’.

• Line 371-398: I would recommend to stick to suggesting regulations related directly to the results of this study. For example, I completely support the recommendation to implement warning labels, but this isn’t directly related to UPFs. In line with this I also completely agree with suggestion (iv), but this isn’t really a policy recommendation that can be made based on the current results (could be if this becomes a more important focus of the discussion, but that currently isn’t the case).

• Line 377: Change to ‘where they have these recommendations:…’ (without capital letter).

• Line 404: Maybe use ‘nonetheless’ instead of ‘however’?

Conclusions

• Overall I would recommend to keep the conclusion shorter and more focussed on the true results of the study itself.

• Line 430-433: In my opinion this would better fit under the discussion as this is not a results of the study itself.

• Line 433: Potentially reformulate to something like ‘These data highlight the need to improve the current health promotion and prevention policies….’.

7. PLOS authors have the option to publish the peer review history of their article (what does this mean?). If published, this will include your full peer review and any attached files.

**Do you want your identity to be public for this peer review?** For information about this choice, including consent withdrawal, please see our Privacy Policy.

Reviewer #1: No

Reviewer #2: **Yes: **Iris Van Dam

---

## [Editor Report · Decision Letter 2]

7 Dec 2023

The increasing trend in the consumption of ultra-processed food products is associated with a diet related to chronic diseases in Colombia. Evidence from National Nutrition Surveys 2005 and 2015.

PGPH-D-23-00294R2

Dear Dr Cediel,

We are pleased to inform you that your manuscript 'The increasing trend in the consumption of ultra-processed food products is associated with a diet related to chronic diseases in Colombia. Evidence from National Nutrition Surveys 2005 and 2015.' has been provisionally accepted for publication in PLOS Global Public Health.

Best regards,

Leonor Guariguata, MPH, PhD

Academic Editor